# Facemask-wearing behavior to prevent COVID-19 and associated factors among public and private bank workers in Ethiopia

**Seada Hassen** [ID]*[o], **Metadel Adane** [ID][o]

Department of Environmental Health, College of Medicine and Health Science, Wollo University, Dessie, Ethiopia

[o] These authors contributed equally to this work.

* seah9400m@gmail.com

## Abstract

### Background

Given widespread vulnerability to COVID-19 infection in areas with low vaccination rates, facemask wearing is repeatedly emphasized for the general population including bank workers, who have contact with many customers each day. Over the first year of the COVID-19 pandemic, studies focused on facemask wearing among healthcare workers but not among bank workers, who are also at risk of COVID-19. To address this gap and to consider intervention measures that encourage the wearing of facemasks, this study was conducted to identify behaviors of facemask wearing and associated factors among bank workers in Dessie City, Ethiopia.

### Materials and methods

An institution-based cross-sectional study was conducted among 413 bank workers who were selected using a simple random sampling technique from January 1st to 30th, 2021 in Dessie City, Ethiopia. Data were collected using a structured questionnaire and on-the-spot observational checklist. The collected data were checked, coded and entered to EpiData version 4.6 and exported to Statistical Package for Social Sciences (SPSS) version 25.0 for data cleaning and analysis. Data were analyzed using bivariable (crude odds ratio [COR]) and multivariable (adjusted odds ratio [AOR]) logistic regression model at 95% confidence interval (CI). Variables from the bivariable analysis with a $p$-value <0.25 were considered for multivariable analysis. Then, variables that had a $p$-value <0.05 were declared as factors significantly associated with behavior of facemask wearing.

### Main findings

In this study, the behavior of facemask wearing among bank workers was 50.4% [95%CI: 45.3–55.2%] with 21.1% always wearing a facemask, 72.4% sometimes, and 6.5% never. A majority of the bank workers 350 (84.7%) had good knowledge of COVID-19 and half of them 208 (50.4%) had a positive attitude towards taking precautions against COVID-19.

**Data Availability Statement:** All relevant data are found in the manuscript and its Supporting Information files.

**Funding:** No funding was received for this work.

**Competing interests:** Authors have no competing interests.

**Abbreviations:** AOR, adjusted odds ratio; CI, confidence interval; COR, crude odds ratio; CDC, communicable disease control and prevention; COVID-19, coronavirus disease 2019; SARS-CoV-2, severe acute respiratory syndrome coronavirus-2; WHO, World Health Organization.

Just over two-thirds of the respondents 284 (68.8%) preferred to wear a non-medical mask. Two hundred fifty-five (61.7%) said wearing a facemask interfered with communication and 259 (62.7%) felt that wearing a facemask was not comfortable. Facemask-wearing behavior was significantly associated with a high level of positive attitude towards taking precautions against COVID-19 (AOR = 3.27, 95% CI: 1.75–6.11), the perception that the consequences of getting COVID-19 could be serious (AOR = 4.87, 95% CI: 2.38–9.94), the presence of chronic illness (AOR = 2.19, 95% CI: 1.07–4.48), sex being female (AOR = 1.87, 95% CI: 1.06–3.32) and age being greater or equal to 35 years (AOR = 9.25, 95% CI: 4.79–17.88).

## Conclusion

The main finding of the study showed that the behavior of facemask wearing among bank workers was relatively low (50.4%) compared to other types of workers as found in other studies. To increase the behavior of facemask wearing among bank workers, health decision makers need to develop updated guidance for promotion of facemask wearing to increase the practical and appropriate use of facemasks among bank workers. Bank managers and concerned government bodies should enforce mask use to change behavior of these workers.

## Introduction

Coronavirus disease 2019 (COVID-19) is an emerging respiratory disease that was first reported in Wuhan, Hubei Province, China, on December 31st, 2019 as a cluster of pneumonia cases. The disease is caused by the novel severe acute respiratory syndrome coronavirus-2 (SARS-CoV-2) [1]. On January 7th, 2020, the etiological agent of the pneumonia was officially announced as a novel coronavirus. Among the coronaviruses that infect humans, severe acute respiratory syndrome coronavirus (SARS-CoV) and Middle East respiratory syndrome-related coronavirus (MERS-CoV) are highly pathogenic [2].

COVID-19 has an incubation period of between 2 and 14 days, with an average of 5 days [3]. After the incubation period, COVID-19 shows a mild course in 80.0% of observed cases and a severe course in 20.0%, with a lethality rate of 0.3–5.8% [4]. The world entered the second year of the COVID-19 pandemic applying preventive measures only, but after over 110 million global infections and 2.4 million deaths, the development of COVID-19 vaccines offered a glimmer of hope, although they have not yet been distributed throughout all countries. Vaccinations help to create herd immunity when vaccine coverage reaches sufficient levels, and they reduce morbidity and mortality induced by the virus [5].

At the onset of the pandemic, public health interventions including facemask wearing, handwashing or sanitizer use, and keeping physical distance were practiced by government enforcement in Dessie City, although the punishment had been abolished by the time the data were gathered. Facemask wearing is available and advisable for preventing transmission pathways for coronavirus by acting as a mechanical barrier [4,6]. Facemask wearing is also used to protect the self against the spreading of droplets from sick to healthy individuals by completely covering the mouth and nose and adjusting well to the face [7,8]. The use of masks is part of a comprehensive package of prevention and control measures that can limit the spread of certain respiratory viral diseases, including COVID-19. Masks can be used either for protection of

healthy persons or to prevent persons having COVID-19 disease from transmitting it to others [9].

Bank workers are among the many service-sector employees who have frequent and close interaction with many people. Many of these workers have either direct or indirect physical contact with the public through exchange of money, which is an exposure route for COVID-19 transmission that is not clearly seen [8,10]. The banking industry's major working tool is cash that is processed by cashiers, which allows the chance of COVID-19 exposure to various individuals [11]. Cases of COVID-19 spread all over the world, increasing rapidly due to the presence of community transmission [12].

Globally, there were 111,518,562 confirmed cases, 2,468,646 deaths and 62,903,843 recoveries as of February 22, 2021 at 9:24 GMT. The first case of COVID-19 on the continent of Africa was reported on February 14[th], 2020. By May 13th, cases had been reported in all 54 African countries [13]. In Ethiopia, the first case of COVID-19 was reported on March 13[th],2020, and the number of reported cases is increasing with improved testing [14,15]. By February 22, 2021, 19:15 GMT, COVID-19 cases rose to 153,541, with deaths of 2,293 and 131,713 recoveries in Ethiopia [16]. COVID-19 cases were also being reported in Dessie City at the time of this study's inquiry.

Bank employees come into contact with the virus by close interaction with clients or coworkers without appropriate physical distancing [10]. Given widespread population vulnerability to COVID-19 infection, facemask wearing is repeatedly emphasized for the whole population including public and private bank workers, but until very recently, studies focused on the importance of facemask wearing in healthcare workers [17] and did not consider bank workers, who are also at risk of COVID-19.

Ensuring that evidence-based preventive measures and practices are consistently applied among public and private bank workers will help to prevent human-to-human transmission of pathogens including SARS-CoV-2 [18]. There had been no studies on the prevalence of and preventive strategies against COVID-19 among bank employees in Dessie City before the start of this study. To address the gap and better understand the current situation of facemask-wearing behavior and its associated factors among public and private bank workers, this study was conducted in Dessie City, Ethiopia.

## Materials and methods

### Study area

The study was conducted in private and public banks located in Dessie City, the capital city of south Wollo Zone, on the eastern margin of Amhara Regional State in the north central part of Ethiopia, 401km from Addis Ababa. According to the 2007 national census conducted by the Central Statistical Agency (CSA), Dessie City's total population was projected at 212,436 for 2014. In the year 2020, there were 17 government and 24 private bank branches having around 2647 workers in Dessie City [19,20].

### Study design, period and population

An institution-based cross-sectional study was conducted to determine facemask-wearing behavior and associated factors among bank workers in Dessie City from January 1[st] to 30[th], 2021. All workers in bank branches of Dessie City were the source population, while selected bank workers from selected branches of banks in Dessie City were the study population.

## Sample size determination and sampling technique

Sample size was determined using single population proportion formula, $n = \frac{(z_{a/2})^2 \,*\, p(1-p)}{d^2}$ considering the assumptions that the proportion of facemask wearing in institutions including banks of Dessie City at 50% (50% was considered since there had been no previous study conducted on bank workers in the study area), a 95% CI and 5% margin of error. After considering a 10% non-response rate from the initial calculated sample size, the final sample size for this study became 422.

There were 17 government and 24 private bank branches in Dessie City, for a total of 41 bank branches, from which 50% (21 bank branches, 9 government and 12 private) were randomly selected. The numbers of workers were taken from attendance sheets in each selected branch. After getting their number, samples were taken randomly from the sheet first by allocating the entire sample proportionally to the total number of workers in the selected bank branches.

## Operational definitions

**Bank workers.**   Both back- and frontline officers in the bank who are responsible for accepting customers' cash deposits and utility payments, recording transactions, printing receipts, cashing cheques and advising customers about investments, foreign currency exchange and loans frequently have contact with customers.

**Facemask-wearing behavior.**   The practice by bank workers of wearing a facemask covering the nose, mouth, and lower jaw at the time of data collection divided by the total number of study participant bank workers.

**High level of positive attitude.**   Those bank workers responding positively to more than or equal to the mean out of 11 attitude questions about taking precautions to prevent COVID-19 transmission.

**Low level of positive attitude.**   Those bank workers responding positively to fewer than the mean out of 11 attitude questions about taking precautions to prevent COVID-19 transmission.

**Good knowledge.**   Those bank workers responding correctly to more than or equal to the mean of a total of 16 knowledge questions.

**Poor knowledge.**   Those bank workers responding correctly to fewer than the mean of a total of 16 knowledge questions.

**Medical masks.**   Surgical or procedure masks that are flat or pleated; and also N95 facemasks affixed to the head with straps that go around the ears or head or both [21,22].

**Non-medical masks.**   Masks made out of different combinations of fabrics (cloth), layering sequences and available in diverse shapes [22].

## Data collection procedures and quality assurance

A structured questionnaire and an observational checklist were developed after reviewing the literature [23–26], a WHO report on COVID-19 [9], and an Ethiopian Ministry of Health report [27]. To keep the questions consistent, the instrument was prepared in English, translated to the local language (Amharic) and then retranslated to English. The first part of the questionnaire asked about facemask-wearing behaviors, scored according to whether or not the respondents wore a facemask at the time of data collection, which is a method similar to other study [28]; the next part included questions about risk factors for those behaviors to be poor or good, including information on socio-demographic and economic factors; bank environment and service-related factors; knowledge and attitude about COVID-19; behavioral

factors; and factors related to medical history and subjects' source of information about COVID-19.

Knowledge and attitudes towards COVID-19 were also probed via 16 and 11 questions, respectively. One point was awarded for each correct answer, while zero points were given for each item that was answered incorrectly or left unanswered by selecting the response 'do not know.' The possible knowledge score ranged from 0 to 16, with a score higher than the mean indicating a better level of knowledge on COVID-19 possessed by the participant [25]. Each item on attitude towards taking precautions against COVID-19 was rated on a 5-point Likert scale ranging from 1 (strongly disagree) to 5 (strongly agree). The mean score of each subscale was calculated to indicate the degree of participant's attitudes in the respective domains [3].

Before the commencement of the actual data collection, the questionnaire was pretested on five other bank branches that had not been selected for the study. The data were collected by self-administration and observation of utilization of facemask and other bank service-related questions. Three data collectors were recruited and given one day of training about study objectives, data collection tools and ethical issues; they distributed the questionnaires and observed the bank environment using the checklists. The investigator and supervisors checked the completeness of the questionnaires on a daily basis for data quality control.

In order to assure the survey tool's validity, the questionnaire was created after analyzing a variety of published literature and reports. Data entry was also reviewed in a randomly selected 10% of the surveys to ensure the questionnaire's reliability, and double data entry was performed to prevent data entry errors. Data cleaning prior to statistical analysis was also carried out.

## Data management and analysis

Data were checked for completeness and consistency, then coded and entered to EpiData version 4.6 and exported to SPSS version 25.0 for data cleaning and analysis. Descriptive analysis was carried out and the results were presented using frequencies with percentages (%) for categorical variables and mean with standard deviations (SD) for continuous variables. Bank workers wearing a facemask at the time of observation were represented with '1' and those not wearing a facemask with '0'. The number of bank workers wearing a mask covering the nose, mouth, and lower jaw at the time of observation divided by the total number of study participants was considered as the proportion of those with "good behavior of facemask wearing"; whereas the proportion of those with 'poor face-mask wearing behavior' was calculated the same way using the number who were not similarly wearing a mask. Study participant wearing of any type of mask as observed during data collection was considered as facemask-wearing behavior in the study.

A binary logistic regression model was used for data analysis at 95% CI; bivariable (crude odds ratio [COR]) analysis was performed and variables with p-value <0.25 were transported to multivariable logistic regression model to identify factors independently associated with the behavior of facemask wearing. Finally, variables with p-value <0.05 in multivariable logistic regression were taken as statistically significant, and adjusted odds ratio (AOR) with 95% CI was considered to see strength and significance of the association, respectively. Multicollinearity test carried out between independent variables where the standard error cut-off point was greater than 2 showed it did not happen. The Hosmer–Lemeshow goodness-of-fit test was used, finding the p-value was 0.938, indicating that the model was fit.

## Ethical considerations

An ethical clearance letter was obtained from the ethical review committee of Wollo University College of Medicine and Health Science. Letters of permission from the Dessie City health

bureau and government and private bank branches were obtained. As per WHO guidelines to prevent COVID-19 transmission, data collectors wore facemasks, used hand sanitizer during distribution and collection of questionnaires, and kept physical distance of two meters (6'8") at the time of data collection. Before distributing the questionnaire, written consent was obtained from participants by attaching one page to the beginning of the questionnaire. Possible identifiers such as names of the participants were not requested to ensure confidentiality. Study participants who did not wear a facemask at the time of interview were advised to wear a facemask if they had one, and a face-mask was provided o those who did not have one during the data collection.

## Results

### Socio-demographic and economic characteristics of bank workers

The response from the total sample of 422 was 413 (97.9%), including 208 (50.4%) government and 205 (49.6%) private bank branches. Of the total respondents, 233 (56.4%) were male, 180 (43.6%) had a first degree and a majority of the respondents 247 (59.8%) were within the age range of 18–34 years, with a mean age of 33.18 years (SD [standard deviation ±8.13 years) (Table 1).

### Bank environment related factors

Almost equal numbers of workers were taken from government (50.4%) and private (49.6%) bank branches for this study. The mean number of staff and daily customers were 23 and 305, respectively (Table 2).

### Knowledge and attitude factors about COVID-19

The majority of respondents 350 (84.7%) had good knowledge about COVID-19 and 63 (15.3%) had poor knowledge about COVID-19, while half of the respondents 208 (50.4%) had a high level of good attitude towards taking precautions against COVID-19 (Table 3).

### Behavioral factors among bank workers

Of the total sample in this study, 50.4% (95% CI: 45.3–55.2) of bank workers showed good mask-wearing behavior and 49.6% (95% CI: 44.8–54.7) showed poor mask-wearing behavior. Over two-thirds of respondents 284 (68.8%) preferred a non-medical mask for protection from inhalation of droplets and air from outside. More than three-fourths 330 (79.9%) of bank workers felt fear of COVID-19 and 245 (59.3%) felt that the consequences of getting COVID-19 could be serious. One-third 138 (33.4%) of the bank workers kept a physical distance from customers. Two hundred fifty-five (61.7%) said wearing a facemask interfered with communication and 259 (62.7%) felt that wearing a facemask was not comfortable (Table 4).

### Medical history-related and COVID-19 source-of-information factors

Less than one-third 112 (27.1%) of bank workers had presence of a respiratory condition and about one-fifth 80 (19.4%) had a chronic illness. From a total of 318 respondents who had received health information about COVID-19, 257 (80.82%) received it from television, radio or newspaper, 112 (35.22%) from health care providers, 213 (66.98%) from social media and 159 (50%) from friends. About three-fourths 300 (74.8%) of bank workers had received training on COVID-19, whereas 104 (25.2%) had not received training about COVID-19. Thirty-seven respondents had taken training about COVID-19 on only one occasion from a total number of respondents of 104 who got the training (Table 5).

**Table 1. Socio-demographic and economic factors and bivariable analysis with facemask-wearing behaviors among bank workers in Dessie City, Ethiopia, January 2021.**

| Variables | Categories | Frequency | Facemask-wearing behavior | | COR (95% CI) | p-value |
|---|---|---|---|---|---|---|
| | | n (%) | Good | Poor | | |
| Sex of respondent | Male | 233 (56.4) | 90 | 143 | 1 | |
| | Female | 180 (43.6) | 118 | 62 | 3.02 (2.02–4.53) | <0.001 |
| Age of respondent (years) | 18–34 | 247 (59.8) | 64 | 183 | 1 | |
| | > = 35 | 166 (40.2) | 144 | 22 | 18.7 (11.0–31.8) | <0.001 |
| Education level | Diploma | 101 (24.5) | 38 | 63 | 1 | |
| | 1st degree | 180 (43.6) | 71 | 109 | 1.08 (0.65–2.83) | 0.764 |
| | 2nd degree | 132 (32.0) | 99 | 33 | 4.97 (2.83–8.74) | <0.001 |
| Monthly income (USD, United States Dollar)* | 127.8–299.0 | 210 (50.8) | 72 | 138 | 1 | |
| | 299.1–689.9 | 203 (49.2) | 136 | 67 | 3.89 (2.59–5.85) | <0.001 |
| Marital status | Not married | 179 (43.3) | 52 | 127 | 1 | |
| | Married | 234 (56.7) | 156 | 78 | 4.89 (3.20–7.45) | <0.001 |
| Experience at the bank (years) | <2 | 71 (17.2) | 22 | 49 | 1 | |
| | 2–5 | 194 (47.0) | 69 | 125 | 1.23 (0.69–2.20) | 0.487 |
| | >5 | 148 (35.8) | 117 | 31 | 8.41 (4.43–15.9) | <0.001 |
| Position as cashier (days/week) | 0–4 | 130 (31.5) | 59 | 71 | 1 | |
| | 5 or 6 | 283 (68.5) | 149 | 134 | 1.34 (0.88–2.03) | 0.171 |
| Family size (persons) | <5 | 245 (59.3) | 89 | 156 | 1 | |
| | >/ = 5 | 168 (40.7) | 119 | 49 | 4.26 (2.79–6.50) | <0.001 |
| Have children in household | No | 196 (47.5) | 78 | 118 | 1 | |
| | Yes | 217 (52.5) | 130 | 87 | 2.26 (1.52–3.35) | <0.001 |
| Have family member >65 years old in household | No | 333 (80.6) | 167 | 166 | 1 | |
| | Yes | 80 (19.4) | 41 | 39 | 1.05 (0.64–1.70) | 0.860 |

1, reference category; COR, crude odds ratio; CI, confidence interval.

*Average exchange of 1 USD (United States Dollar) to ETB (Ethiopian Birr) was 39.13224 during January 2021.

## Multivariable logistic regression analysis

From a total of 28 variables with p<0.25 that were entered to multivariable logistic regression (MLR) analysis model, five variables were found to have significant and independent association with facemask-wearing behavior. The main findings of this study showed that female respondents were 1.87 times (AOR = 1.87; 95%CI: 1.06–3.32) more likely to wear a facemask

**Table 2. Bank environment-related factors and bivariable analysis with facemask-wearing behaviors among bank workers in Dessie City, Ethiopia, January 2021.**

| Variables | Categories | Frequency | Facemask-wearing behavior | | COR (95% CI) | p-value |
|---|---|---|---|---|---|---|
| | | n (%) | Good | Poor | | |
| Bank branch type | Government | 208 (50.4) | 118 | 90 | 1.68 (1.14–2.47) | 0.009 |
| | Private | 205 (49.6) | 90 | 115 | 1 | |
| No. of staff in the bank | <23 | 234 (56.7) | 121 | 113 | 1 | |
| | > = 23 | 179 (43.3) | 87 | 92 | 0.88 (0.59–1.30) | 0.532 |
| No. of customers of the bank | 8–304 | 272 (65.9) | 124 | 148 | 1 | |
| | 305–1,400 | 141 (34.1) | 84 | 57 | 1.76 (1.17–2.67) | 0.007 |

1, reference category; COR, crude odds ratio; CI, confidence interval.

**Table 3. Knowledge- and attitude-related factors and bivariable analysis with facemask-wearing behaviors among bank workers in Dessie City, Ethiopia, January 2021.**

| Variable | Categories | Frequency | Facemask-wearing behavior | | COR (95% CI) | *p*-value |
|---|---|---|---|---|---|---|
| | | *n* (%) | Good | Poor | | |
| **Knowledge about COVID-19** | Poor knowledge | 63 (15.3) | 16 | 47 | 1 | |
| | Good knowledge | 350 (84.7) | 192 | 158 | 3.57 (1.95–6.54) | <0.001 |
| **Attitude towards taking precautions against COVID-19** | Low level of positive attitude | 205 (49.6) | 46 | 159 | 1 | |
| | High level of positive attitude | 208 (50.4) | 162 | 46 | 12.2 (7.66–19.4) | <0.001 |

1, reference category; COR, crude odds ratio; CI, confidence interval.

compared to male workers, those 35 years of age or over were 9.25 times (AOR = 9.25; 95%CI: 4.79–17.88) more likely to wear a facemask than those who were 18–34. Those having a high level of positive attitude towards taking precautions against COVID-19 were 3.27 times

**Table 4. Behavioral factors and bivariable analysis with facemask-wearing behaviors among bank workers in Dessie City, Ethiopia, January 2021.**

| Variables | Categories | Frequency | Facemask-wearing behavior | | COR (95% CI) | *p*-value |
|---|---|---|---|---|---|---|
| | | *n* (%) | Good | Poor | | |
| **Type of mask** | Medical | 129 (31.2) | 70 | 59 | 1 | |
| | Non-medical | 284 (68.8) | 138 | 146 | 0.80 (0.53–1.21) | 0.286 |
| **Feel vulnerable to contracting COVID-19** | No | 216 (52.3) | 60 | 156 | 1 | |
| | Yes | 197 (47.7) | 148 | 49 | 7.85 (5.06–12.19) | <0.001 |
| **Feel fear of COVID-19** | No | 83 (20.1) | 27 | 56 | 1 | |
| | Yes | 330 (79.9) | 181 | 149 | 2.52 (1.52–4.19) | <0.001 |
| **Feel that the consequences of getting COVID-19 could be serious** | No | 168 (40.7) | 23 | 145 | 1 | |
| | Yes | 245 (59.3) | 185 | 60 | 19.44 (11.47–32.94) | <0.001 |
| **Know someone who had positive test results for SARS-CoV-2** | No | 207 (50.1) | 64 | 143 | 1 | |
| | Yes | 206 (49.9) | 144 | 62 | 5.19 (3.41–7.89) | <0.001 |
| **Know someone who died from COVID-19** | No | 231 (55.9) | 74 | 157 | 1 | |
| | Yes | 182 (44.1) | 134 | 48 | 5.92 (3.85–9.12) | <0.001 |
| **Travel outside the country** | No | 381 (92.3) | 189 | 192 | 1 | |
| | Yes | 32 (7.7) | 19 | 13 | 1.49 (0.71–3.09) | 0.291 |
| **Money-counting techniques** | Using saliva | 54 (13.1) | 17 | 37 | 1 | |
| | Using chemical | 188 (45.5) | 101 | 87 | 2.53 (1.33–4.80) | 0.005 |
| | Using water | 75 (18.2) | 45 | 30 | 3.27 (1.56–6.82) | 0.002 |
| | Don't use any liquid | 96 (23.2) | 45 | 51 | 1.92 (0.95–3.87) | 0.068 |
| **Keep physical distance from coworkers** | No | 275 (66.6) | 130 | 145 | 1 | |
| | Yes | 138 (33.4) | 78 | 60 | 1.45 (0.96–2.19) | 0.077 |
| **Keep physical distance from customers** | No | 271 (65.6) | 130 | 141 | 1 | |
| | Yes | 142 (34.4) | 78 | 64 | 1.32 (0.88–1.99) | 0.180 |
| **Comfortable wearing facemask** | No | 259 (62.7) | 80 | 179 | 1 | |
| | Yes | 154 (37.3) | 128 | 26 | 11.02 (6.70–18.11) | <0.001 |
| **Communication interference caused by facemask** | No | 158 (38.3) | 125 | 33 | 7.85 (4.93–12.49) | <0.001 |
| | Yes | 255 (61.7) | 83 | 172 | 1 | |
| **Facemask leaves marks on face** | No | 277 (67.1) | 143 | 134 | 1.17 (0.77–1.76) | 0.465 |
| | Yes | 136 (32.9) | 65 | 71 | 1 | |

1, reference category; COR, crude odds ratio; CI, confidence interval.

**Table 5. Medical history-related and COVID-19 source-of-information factors and bivariable analysis with facemask-wearing behaviors among bank workers in Dessie City, Ethiopia, January 2021.**

| Variables | Categories | Frequency | Facemask-wearing behavior | | COR (95% CI) | p-value |
|---|---|---|---|---|---|---|
| | | n (%) | Good | Poor | | |
| **Presence of respiratory condition** | No | 301 (72.9) | 145 | 156 | 1 | |
| | Yes | 112 (27.1) | 63 | 49 | 1.38 (0.89–2.14) | 0.145 |
| **Presence of chronic illness** | No | 333 (80.6) | 155 | 178 | 1 | |
| | Yes | 80 (19.4) | 53 | 27 | 2.25 (1.35–3.76) | 0.002 |
| **Training on COVID-19** | No | 309 (74.8) | 125 | 184 | 1 | |
| | Yes* | 104 (25.2) | 83 | 21 | 5.82 (3.43–9.88) | <0.001 |
| **Given health information on COVID-19** | No | 95 (23.0) | 20 | 75 | 1 | |
| | Yes** | 318 (77.0) | 188 | 130 | 5.42 (3.16–9.32) | <0.001 |

1, reference category; COR, crude odds ratio; CI, confidence interval.

*Thirty-seven respondents had taken training about COVID-19 on only one occasion from a total number of respondents of 104 who got the training.

**257 (80.82%) received it from television, radio and newspaper, 112 (35.22%) from health care providers, 213 (66.98%) from social media and 159 (50%) from friends.

(AOR = 3.27; 95%CI: 1.75–6.11) more likely to wear a facemask compared to those having a low level of positive attitude towards taking precautions against COVID-19 (Table 6).

Furthermore, bank workers who felt that the consequences of getting COVID-19 could be serious were 4.87 times (AOR = 4.87; 95%CI: 2.38–9.94) more likely to wear a facemask compared to those who didn't feel that the consequences of getting COVID-19 could be serious, and the odds of wearing a facemask among those participants having a chronic illness were 2.19 times (AOR = 2.19; 95%CI: 1.07–4.48) higher than those who did not have a chronic illness (Table 6).

**Table 6. Factors associated with facemask-wearing behavior from multivariable logistic regression analysis among bank workers in Dessie City, Ethiopia, January 2021.**

| Variables | Categories | Facemask-wearing behavior | | AOR (95% CI) | p-value |
|---|---|---|---|---|---|
| | | Good | Poor | | |
| **Sex of respondent** | Male | 90 | 143 | 1 | |
| | Female | 118 | 62 | 1.87 (1.06–3.32) | 0.031 |
| **Age of respondent (years)** | 18–34 | 64 | 183 | 1 | |
| | > = 35 | 144 | 22 | 9.25 (4.79–17.88) | <0.001 |
| **Attitude towards taking precautions against COVID-19** | Low level of positive attitude | 46 | 159 | 1 | |
| | High level of positive attitude | 162 | 46 | 3.27 (1.75–6.11) | <0.001 |
| **Feel that the consequences of getting COVID-19 could be serious** | No | 23 | 145 | 1 | |
| | Yes | 185 | 60 | 4.87 (2.38–9.94) | <0.001 |
| **Know someone who had positive test results for SARS-CoV-2** | No | 64 | 143 | 1 | |
| | Yes | 144 | 62 | 1.91 (0.96–3.80) | 0.067 |
| **Know someone who died from COVID-19** | No | 74 | 157 | 1 | |
| | Yes | 134 | 48 | 0.48 (0.22–1.04) | 0.063 |
| **Presence of chronic illness** | No | 155 | 178 | 1 | |
| | Yes | 53 | 27 | 2.19 (1.07–4.48) | 0.032 |

AOR, adjusted odds ratio; CI, confidence interval; 1, reference category.

## Discussion

In this institution-based cross-sectional study conducted to determine factors related to the behavior of facemask wearing among public and private bank workers of Dessie City, it was found that 50.4% of respondents had good facemask-wearing behavior and that the behavior of facemask wearing among bank workers was significantly associated with female sex, age being > = 35 years, having a high level of positive attitude towards taking precautions against COVID-19, feeling that the consequences of getting COVID-19 could be serious, and presence of chronic illness.

Facemask-wearing behavior among bank workers in this study was 50.4%. This result is similar to the findings of a study among primary school students in Wuhan, China 51.6% [28], 50.0% found in study by Barasheed et al. [29], and 54.68% found in a study done on taxi drivers in Dessie City and Kombolcha Town [30]. This similar result of studies on facemask-wearing behavior may be due to similarity of study sources being from institutions.

Facemask-wearing behavior was found to be lower in this study than in a study done in Brazil where it was 95.5% [7], in Nigeria where it was 64.5% [31], and in Hong Kong where wearing masks in public was found to be 94.3% [32]. In a South Korean study, 63.2% reported always wearing a face mask when outside [33] while 97.9% used a facemask in a community in China [34]. Overall, 61.2% of respondents reported consistent use of a facemask to prevent SARS in Hong Kong [35] and in China, nearly all of the participants (98.0%) wore a facemask when going out [25]. Mask mandates enacted in the United States of America in late July and August increased mask-wearing compliance to over 90% in all groups of population [36]. The reason for facemask wearing to be lower in our study may be due to the fading of government pressure that occurred early in the pandemic to wear a facemask and the ceasing of punishment of those who did not. The difference may also be due to factors such as different study area and study period.

But the behavior of facemask wearing in this study was higher than found in a study in Japan where it was 38% [37], and in Ghana where 31.5% of the students wore a facemask often or always [38]. This may be due to a lack of action being taken to improve rates in areas where facemask-wearing behavior is low. And individuals may not understand that facemask use could result in a large reduction in risk of infection.

In our study, bank workers who were female had 1.87 times greater chance of wearing a facemask. There are similar studies showing that being a woman increased the likelihood of wearing facemasks in Wuhan, China ($p<0.001$) (29), and in Brazil [7]. Women were more likely to practice these behaviors than men ($p<0.001$) in other studies also [35,36]. The odds of an individual wearing a mask increased significantly with female sex and were 1.5-times greater for females than males, according to a study in America [36]. Participants who were male were less likely to implement preventive measures including facemask wearing in Hong Kong [33]. This may be because women worry more than men about COVID-19 disease for themselves, their families and for individuals with whom they have contact.

This study shows that respondents whose age was > = 35 years were 9.25 times more likely to have good facemask-wearing behavior. Older adults are likely to have better COVID-19 preventive practices than younger people [39]. The odds of an individual wearing a mask increased significantly with age [7]. And those in an older age group of 50–59 years were more likely to wear a facemask [35]. Older individuals are more highly vulnerable to getting a serious case of COVID-19 compared to younger people, which may be the reason for older people protecting themselves from exposure by having good facemask-wearing behavior.

In this study, those with a perception of serious consequences of getting COVID-19 had 4.87 times higher chance of practicing facemask-wearing behavior. A study done on

perception and practices during the COVID-19 pandemic in an urban community in Nigeria revealed that a perception of the likelihood of contracting COVID-19 was a factor for applying COVID-19 preventive measures [31]. From a study conducted on barriers to mask wearing for influenza-like illnesses among urban Hispanic households, the perception of the risk of disease was one factor [40].

Perceived fatality, efficacy of wearing facemasks, and mental distress because of influenza A/H1N1 were associated with facemask use in public areas (AOR, 1.53 to 2.52) [41]. Correlation results indicated that perceived risk was related to preventive behaviors like facemask wearing [42]. Participants who perceived a lower risk of being infected were less likely to implement preventive measures such as facemask wearing [32]. This perception of risk of getting the disease became a factor for wearing facemask due to the fact that a person who perceives risk tries to use preventive techniques to protect him- or herself.

This study also shows that bank workers with a chronic illness were 2.19 times more likely to wear a facemask. People with chronic diseases are important populations to consider with respect to the current wide-spread recommendation for facemask use; for example, wearing either a half-mask respirator or an N95 mask among individuals with mild pulmonary diseases including asthma, chronic rhinitis, and chronic obstructive pulmonary disease (COPD) may be considered [43].

Based on the wide range of chronic diseases and severity of each person's disease, the decision to wear a mask will likely need to be made individually and with consultation from a physician, given the individual's particular circumstances. Individuals with pre-existing chronic diseases such as diabetes, hypertension and obesity (metabolic syndrome) are at an increased risk of hospitalization and mortality with COVID-19 [44]. This underscores the importance of wearing masks to help protect this vulnerable population. That said, if an individual with a chronic disease is unable to safely wear a mask, the responsibility may fall to other healthy individuals to wear masks to protect the vulnerable [43].

## Limitations of the study

Limitations of the study include the fact that we did not observe the home environment of the workers, which limited our ability to examine COVID-19 prevention in their living environment. In addition to this, the scarcity of COVID-19-related studies on the behavior of facemask wearing among bank workers forced us to use other studies conducted in different source populations at institutions and at community levels, which reduces the strength of the discussion.

The data collection being self-administered for the sake of COVID-19 prevention and the short time workers had to respond might have biased the self-reported data. Besides these limitations, overall, this study can provide appropriate information about bank workers in Dessie City regarding facemask-wearing behavior, especially at their work sites where they have contact with many people while providing service for customers. It also supports the need for measures to help bank workers wear facemasks and for bank managers to enforce the wearing of facemasks to prevent COVID-19 transmission.

## Conclusion

The main finding of the study was that the behavior of facemask wearing among bank workers was relatively low at 50.4%, and the factors significantly associated with good facemask-wearing behavior among the bank workers were sex, age, attitude towards taking precautions against COVID-19, perception towards getting COVID-19 (felt that the consequences of getting COVID-19 could be serious) and chronic illness.

We recommended that concerned individuals should play their part to increase behavior of facemask wearing among bank workers to minimize the spread of COVID-19. This includes health decision makers who should develop updated guidance for bank workers on the use of facemasks; bank managers who are responsible to address different types of facemasks in reducing both inhaled and surface transmission, to change the behavior of workers towards COVID-19 to reduce the risk of transmission and to enforce wearing of facemasks by workers.

Researchers are recommended to use other strong research designs to perform further study on what control measures are likely to be most effective both to protect workers and to prevent workers spreading disease in the workplace, including banks, and also to protect population as a whole, since bank workers are part of the community; although some variables apply only to bank workers, many of the variables addressed by this study may apply to the general population also.

## Supporting information

**S1 File. English version of the questionnaire.**
(DOCX)

**S2 File. Amharic (local language) version of the questionnaire.**
(DOCX)

**S1 Dataset.**
(XLSX)

## Acknowledgments

We gratefully acknowledge the College of Medicine and Health Science Ethical Review Committee of Wollo University for providing the ethical clearance letter, which allowed us to do this research. We express our special thanks to Dessie City health bureau and bank branch managers for giving permission to do this study. Our appreciation also goes to data collectors, supervisors and study participants for their cooperation during study data collection. Lisa Penttila is highly acknowledged for language editing of the manuscript.

## Author Contributions

**Conceptualization:** Seada Hassen, Metadel Adane.

**Data curation:** Seada Hassen, Metadel Adane.

**Formal analysis:** Seada Hassen, Metadel Adane.

**Investigation:** Seada Hassen, Metadel Adane.

**Methodology:** Seada Hassen, Metadel Adane.

**Project administration:** Seada Hassen, Metadel Adane.

**Resources:** Seada Hassen, Metadel Adane.

**Software:** Seada Hassen, Metadel Adane.

**Supervision:** Seada Hassen, Metadel Adane.

**Validation:** Seada Hassen, Metadel Adane.

**Visualization:** Seada Hassen, Metadel Adane.

**Writing – original draft:** Seada Hassen, Metadel Adane.

**Writing – review & editing:** Seada Hassen, Metadel Adane.

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
