## [Decision Letter · Decision Letter 0]

15 Jun 2021

PONE-D-21-11265

Facemask wearing behavior to prevent COVID-19 and associated factors among bank workers in Dessie City, Ethiopia

PLOS ONE

Dear Dr. Hassen,

Thank you for submitting your manuscript to PLOS ONE. After careful consideration, we feel that it has merit but does not fully meet PLOS ONE’s publication criteria as it currently stands. Therefore, we invite you to submit a revised version of the manuscript that addresses the points raised during the review process.

We look forward to receiving your revised manuscript.

Kind regards,

Zixin Wang, PhD.

Academic Editor

PLOS ONE

Journal Requirements:

2.  Please amend your current ethics statement to address the following concerns: Please explain why written consent was not obtained, how you recorded/documented participant consent, and if the ethics committees/IRBs approved this consent procedure."

3. Please include additional information regarding the survey or questionnaire used in the study and ensure that you have provided sufficient details that others could replicate the analyses. For instance, if you developed a questionnaire as part of this study and it is not under a copyright more restrictive than CC-BY, please include a copy, in both the original language and English, as Supporting Information.  If the original language is written in non-Latin characters, for example Amharic, Chinese, or Korean, please use a file format that ensures these characters are visible."

4. Please state whether you validated the questionnaire prior to testing on study participants. Please provide details regarding the validation group within the methods section.

Reviewers' comments:

Reviewer's Responses to Questions

**Comments to the Author**

1. Is the manuscript technically sound, and do the data support the conclusions?

Reviewer #1: Yes

Reviewer #2: Partly

2. Has the statistical analysis been performed appropriately and rigorously? 

Reviewer #1: Yes

Reviewer #2: No

3. Have the authors made all data underlying the findings in their manuscript fully available?

Reviewer #1: Yes

Reviewer #2: Yes

4. Is the manuscript presented in an intelligible fashion and written in standard English?

Reviewer #1: Yes

Reviewer #2: Yes

5. Review Comments to the Author

Reviewer #1: 1. P4, line 76: there should be a period after the citation.

2. P5, line 89: I suggest to combine the paragraph beginning with “cases of COVID-19” with the first paragraph as these two paragraphs were more closely related, which introduced the situation of COVID-19 pandemic

3. The introduction should provide information about the preventive measures in the Dessie City. Many countries have implemented massive preventive measures to contain the spread of COVID-19 such as lockdown, social distancing, and face-masking wearing in all public venues. Any preventive measures in the investigated city? Is it mandatory? Such background information should be introduced to help readers understand the situation.

4. Any similar COVID-19 research among bank workers (e.g., the prevalence of COVID-19 infection or prevalence of preventive measures in this population)? I think the introduction should give more discussion about this specific population.

5. P6, line 109: as this study was only conducted in the Dessie city, I suggest to describe the COVID-19 situation in Dessie city during the time of investigation.

6. P7, line 31: what about the proportion of randomly selected participants from each back?

7. The author should provide more information about the measurements, e.g., the reliability of the used scales and whether these scales have been used in previous studies or they were self-constructed items. I suggest to add an appendix about the measurements as it was only briefly mentioned in the text.

8. P13, line 277: there should be a period after the citation.

9. Did the participants know they would have a survey about face-masking in advance? This might have an influence on their face-masking behavior due to social desirability. The author may discuss this point.

10. There are some spelling errors (e.g., p12 line 252, it should be "showed"). The author should have a careful check.

11. For discussion, the author should consider the local context and local COVID-19 situation to discuss and interpret the findings as the study was conducted in a single city. The generability of findings should also be discussed.

Reviewer #2: This is an interesting study exploring the associated factors of mask wearing behaviours among bank workers. However, substantial revision is suggested. My major comments are listed below:

1. It is suggested further elaborating on the high risk of COVID-19 infection/transmission and thus needs for targeted health promotion among this population. Now the authors only mentioned that these workers have frequent direct or indirect contact with the public, however, this also applies to many other types of workers in the service industry. It is not clear how the findings would contribute to the extant literature.

2. Although the study focused on a special population group, the studied factors are quite general. To inform the development of tailored promotion strategies among this population, more bank worker-specific factors should be included, such as some contextual factors regarding protection measures provided in the working environment. It shall be mentioned somewhere in the Discussion that how the identified relationships are different from/consistent with those found among general population or other working population.

3. Now the measures of knowledge and attitudes were dichotomized for analysis based on the mean score of the sample rather than a more valid and meaningful cut-off point. I suggest treating them as the original continuous variables as dichotomization itself caused loss of information and reporting the reliability alpha for the attitude scale as well. Also, it is not suggested defining those with higher scores as “positive attitudes” and those with lower scores as “negative attitudes”, as the valence of attitude shall be inherently determined by the measurement items, for example, there are both positive and negative attitudes toward mask wearing.

4. The study explored the use of “non-medical mask” which is good, but please clarify a bit whether the measure of “facemask wearing behaviour” captured this aspect when defining good behaviour of facemask wearing. Please also clarify that whether this behaviour was self-reported or objectively observed by the interview fieldworkers.

5. The study used “cluster random sampling” but not “simple random sampling”, please revise the related parts accordingly. The selected banks shall be included as a covariate in the analysis or multilevel regression models shall be used for handling such clustered data.

6. Table 1-5 shall be combined into one table to make the presentation more succinct.

6. PLOS authors have the option to publish the peer review history of their article (what does this mean?). If published, this will include your full peer review and any attached files.

Reviewer #1: No

Reviewer #2: No

---

## [Author Response · Author response to Decision Letter 0]

13 Jul 2021

Line-by-line response to reviewers 

We would like to express our gratitude to the academic editor and both reviewers for their comments and ideas. We revised the manuscript and our responses are here below. 

For journal office requirements 

Response: - Thank you for this key comment. We formatted the manuscript according to PLOSE ONE author’s guidelines.

2. Please amend your current ethics statement to address the following concerns: Please explain why written consent was not obtained, how you recorded/documented participant consent, and if the ethics committees/IRBs approved this consent procedure."

Response: - Sorry for the confusion we created during writing the ethical statement. We already used written consent based on the decisions of ethical review board committee of Wollo University College of Medicine and Health Science. 

3. Please include additional information regarding the survey or questionnaire used in the study and ensure that you have provided sufficient details that others could replicate the analyses. For instance, if you developed a questionnaire as part of this study and it is not under a copyright more restrictive than CC-BY, please include a copy, in both the original language and English, as Supporting Information. If the original language is written in non-Latin characters, for example Amharic, Chinese, or Korean, please use a file format that ensures these characters are visible."

Response: - Thank you for these key comments. As supporting information, we provided the questionnaire by original language and local language version.

4. Please state whether you validated the questionnaire prior to testing on study participants. Please provide details regarding the validation group within the methods section.

Response: - As discussed in the data quality assurance to keep the validity of the questionnaire, we developed the questionnaire from different published literature and COVID-19 related reports to make it more available. We did a pre-test for the questionnaire before data collection to improve the content of the questionnaire (Please see the revised version in page 9 from lines 186-189).

Response to reviewer #1

1. P4, line 76: there should be a period after the citation.

Response: - Sorry for this error. We accepted the comment and it’s corrected. 

2. P5, line 89: I suggest to combine the paragraph beginning with “cases of COVID-19” with the first paragraph as these two paragraphs were more closely related, which introduced the situation of COVID-19 pandemic.

Response: Thank you and we updated accordingly. (You can see lines 93 for the improvement.)

3. The introduction should provide information about the preventive measures in the Dessie City. Many countries have implemented massive preventive measures to contain the spread of COVID-19 such as lockdown, social distancing, and face-masking wearing in all public venues. Any preventive measures in the investigated city? Is it mandatory? Such background information should be introduced to help readers understand the situation.

Response: Thank you for the very important comments. We included those preventive measures in the introductory part. (Please see in page 4 from lines 77-79 for the improvement.)

4. Any similar COVID-19 research among bank workers (e.g., the prevalence of COVID-19 infection or prevalence of preventive measures in this population)? I think the introduction should give more discussion about this specific population.

Response: Although we appreciate the comments, we are unable to find out any study on prevalence and preventive measures of COVID-19 in this study population before this study is conducted. We mentioned this gap in the background section of the last paragraphs (from line 109-111) and that is way this study was conducted. 

5. P6, line 109: as this study was only conducted in the Dessie city, I suggest to describe the COVID-19 situation in Dessie city during the time of investigation.

Response: - Thank you; we described the situation of COVID-19 in Dessie city. (please see line 100 and 101 of page 5)

6. P7, line 31: what about the proportion of randomly selected participants from each back?

Response: - The numbers of workers were taken from attendance sheets in each selected branch by proportionally determining how much samples to be taken from each branch. After getting their number, samples were taken randomly from the sheet first by allocating the entire sample proportionally to the total number of workers in the selected bank branches using proportional allocation. 

7. The author should provide more information about the measurements, e.g., the reliability of the used scales and whether these scales have been used in previous studies or they were self-constructed items. I suggest adding an appendix about the measurements as it was only briefly mentioned in the text. 

Response: - we tried to elaborate the reliability of the questionnaire as well as the scales used in line with previous used studies as a reference in ‘Data collection procedures and quality assurance ’ part of the manuscript.

8. P13, line 277: there should be a period after the citation.

Response: - corrected

9. Did the participants know they would have a survey about face-masking in advance? This might have an influence on their face-masking behavior due to social desirability. The author may discuss this point. 

Response: - They are told about the objective of the study therefore they know about face-masking since one objective is addressing face-mask wearing behavior of respondents, but we minimize social desirability bias by observing utilization of facemask at the time of data collection. 

10. There are some spelling errors (e.g., p12 line 252, it should be "showed"). The author should have a careful check.

Response: - We corrected all these spelling errors 

11. For discussion, the author should consider the local context and local COVID-19 situation to discuss and interpret the findings as the study was conducted in a single city. The generality of findings should also be discussed. 

Response: - The reason for using COVID-19 situation at global level and study area other than bank was scarcity of COVID-19 related studies on the behavior of facemask wearing among bank workers at local level. 

For reviewer #2

1. It is suggested further elaborating on the high risk of COVID-19 infection/transmission and thus needs for targeted health promotion among this population. Now the authors only mentioned that these workers have frequent direct or indirect contact with the public, however, this also applies to many other types of workers in the service industry. It is not clear how the findings would contribute to the extant literature.

Response: - Starting from the title the target of the study is on bank workers even if there are other sectors that have direct or indirect contact with many peoples. Additional reasons were also included in the manuscript.

2. Although the study focused on a special population group, the studied factors are quite general. To inform the development of tailored promotion strategies among this population, more bank worker-specific factors should be included, such as some contextual factors regarding protection measures provided in the working environment. It

shall be mentioned somewhere in the Discussion that how the identified relationships are different from/consistent with those found among general population or other working population. 

Response: - Despite the fact that general factors were included in the research, this study also included specific factors to analyze the bank environment and bank personnel.

3. Now the measures of knowledge and attitudes were dichotomized for analysis based on the mean score of the sample rather than a more valid and meaningful cut-off point. I suggest treating them as the original continuous variables as dichotomization itself caused loss of information and reporting the reliability alpha for the attitude scale

as well. Also, it is not suggested defining those with higher scores as “positive attitudes” and those with lower scores as “negative attitudes”, as the valence of attitude shall be inherently determined by the measurement items, for example, there are both positive and negative attitudes toward mask wearing. 

Response: - The reason for using mean for categorizing continuous variables of knowledge and attitude is to get appropriate number of responses. And this method was used by other several published COVID-19 papers. The other reason for using the mean is when other way of classification was applied we faced less observation which can’t allow as for further analysis.

4. The study explored the use of “non-medical mask” which is good, but please clarifies a bit whether the measure of “facemask wearing behavior” captured this aspect when defining good behavior of facemask wearing. Please also clarify that whether this behavior was self-reported or objectively observed by the interview fieldworkers. 

Response: - Our criteria for saying good/poor behaviors of face mask wearing; not include type of face mask used as operationally defined, but this descriptive data were collected based on self-reported data since there were respondents who don’t were face mask at the time of data collection. 

5. The study used “cluster random sampling” but not “simple random sampling”, please revise the related parts accordingly. The selected banks shall be included as a covariate in the analysis or multilevel regression models shall be used for handling such clustered data. 

Response: -There was no clustering applied in this study, the only reason for saying government and private bank was to describe and see the difference if any among the two. Like other variables used which were expected to be a factor or not. Simply we select 50% of the bank branches randomly either from government or private and the workers were selected by proportionally allocating their number in that selected branch. 

6. Table 1-5 shall be combined into one table to make the presentation more succinct. 

Response: -If we merge the 5 tables as one Table, it becomes very large and bulky to understand it. 

We would be happy to make further corrections if necessary and look forward to hearing from you all soon. 

I hope that the revised manuscript is accepted for publication in PLoS ONE.

---

## [Decision Letter · Decision Letter 1]

25 Aug 2021

PONE-D-21-11265R1

Facemask wearing behavior to prevent COVID-19 and associated factors among bank workers in Dessie City, Ethiopia

PLOS ONE

Dear Dr. Hassen,

Thank you for submitting your manuscript to PLOS ONE. After careful consideration, we feel that it has merit but does not fully meet PLOS ONE’s publication criteria as it currently stands. Therefore, we invite you to submit a revised version of the manuscript that addresses the points raised during the review process.

We look forward to receiving your revised manuscript.

Kind regards,

Zixin Wang, PhD.

Academic Editor

PLOS ONE

Journal Requirements:

Additional Editor Comments (if provided):

I noticed that some comments raised by Reviewer 2 were not addressed during the revision. The authors should address all comments raised by the reviewers.

Reviewers' comments:

Reviewer's Responses to Questions

**Comments to the Author**

1. If the authors have adequately addressed your comments raised in a previous round of review and you feel that this manuscript is now acceptable for publication, you may indicate that here to bypass the “Comments to the Author” section, enter your conflict of interest statement in the “Confidential to Editor” section, and submit your "Accept" recommendation.

Reviewer #1: All comments have been addressed

Reviewer #2: (No Response)

2. Is the manuscript technically sound, and do the data support the conclusions?

Reviewer #1: Yes

Reviewer #2: Yes

3. Has the statistical analysis been performed appropriately and rigorously? 

Reviewer #1: Yes

Reviewer #2: Yes

4. Have the authors made all data underlying the findings in their manuscript fully available?

Reviewer #1: Yes

Reviewer #2: Yes

5. Is the manuscript presented in an intelligible fashion and written in standard English?

Reviewer #1: Yes

Reviewer #2: Yes

6. Review Comments to the Author

Reviewer #1: (No Response)

Reviewer #2: The authors have addressed some of my previous comments, but the others are left unsolved.

First, please discuss the generalizability of the findings to general population and other work populations. Please refer to my previous #2 comment.

Second, regarding my previous #3 comment, 1) I suggest not defining those with higher scores as “positive attitudes” and and those with lower scores as “negative attitudes”. Instead, high/low level of attitudes can be used; 2) Please report the reliability alpha for the attitude scale. This is necessary even if a dichotomized form was used for analysis.

Third, regarding my previous #4 comment, necessary information has been provided by the authors, please also include it in the manuscript, e.g., the measure of facemask wearing behavior” captured whether wearing a facemask of any type; and it is a self-reported measure.

7. PLOS authors have the option to publish the peer review history of their article (what does this mean?). If published, this will include your full peer review and any attached files.

Reviewer #1: **Yes: **Rui She

Reviewer #2: No

---

## [Author Response · Author response to Decision Letter 1]

15 Sep 2021

Response to editor and reviewer #2

We would like to express our gratitude to the academic editor and both reviewers for their comments and ideas for the first revision. We revised the manuscript for the second time and our responses are here below. 

Additional editor comments 

1. “I noticed that some comments raised by reviewer 2 were not addresses during the revision. The author should address all comments raised by the reviewers”

Response: - Thank you for this main comment for second revision. We see all the comments that were not seen.

For reviewer #2

1. First, please discuss the generalizability of the findings to general population and other work population. Please refer to my previous #2 comment.

Response: - Since the study includes both variables for the general population and specific variables for bank workers we can show the generalizability of the study can be for bank workers as well as for the general population. This generalizability of the study was indicated in the manuscript, line 375-378 of page 17. 

2. Second, regarding my previous #3 comment, 1) I suggest not defining those with higher scores as “positive attitude” and those with lower scores as “negative attitude”. Instead, high/low level of attitudes can be used; 2) please report the reliability alpha for the attitude scale. This is necessary even if a dichotomized form was used for analysis. 

Response: - It was corrected as high/low level of attitude, changed in all parts in the manuscript and Cronbach alpha or reliability alpha coefficient for attitude was 0.87.

3. Third, regarding my previous #4 comment, necessary information has been provided by the authors, please also include it in the manuscript, e.g., the measure of facemask wearing behavior” captured whether wearing a facemask of any type; and it is a self-reported measure. 

Response: - Thank you for all your comments and suggestion, it was indicated in line 197-204, page 9&10 of the manuscript.

We would be happy to make further corrections if necessary and look forward to hearing from you all soon. 

I hope that the revised manuscript is accepted for publication in PLoS ONE.

---

## [Editor Report · Decision Letter 2]

25 Oct 2021

Facemask wearing behavior to prevent COVID-19 and associated factors among bank workers in Dessie City, Ethiopia

PONE-D-21-11265R2

Dear Dr. Hassen,

We’re pleased to inform you that your manuscript has been judged scientifically suitable for publication and will be formally accepted for publication once it meets all outstanding technical requirements.

Kind regards,

Zixin Wang, PhD.

Academic Editor

PLOS ONE
---

## [Editor Report · Acceptance letter]

19 Nov 2021

PONE-D-21-11265R2 

Facemask-wearing behavior to prevent COVID-19 and associated factors among Public and Private Bank Workers in Ethiopia 

Dear Dr. Hassen:

I'm pleased to inform you that your manuscript has been deemed suitable for publication in PLOS ONE. Congratulations! Your manuscript is now with our production department. 

Kind regards, 

on behalf of

Professor Zixin Wang 

Academic Editor

PLOS ONE